

# Mapping sea ice concentration using Nimbus-5 ESMR and local dynamical tie points

Emil Haaber Tellefsen[1,*], Rasmus Tage Tonboe[1,*], and Wiebke Margitta  Kolbe[2]

[1]DTU Space, Orsted Plads building 348, DK-2800 Kgs. Lyngby, Denmark
[2]Danish Meteorological Institute (DMI), National Centre for Climate Research (NCKF), Copenhagen, Denmark
[*]These authors contributed equally to this work.

**Correspondence:** Rasmus Tage Tonboe (rtt@space.dtu.dk)

**Abstract.** As part of the European Space Agency's Climate Change Initiative the one channel US satellite microwave radiometer Nimbus-5 ESMR (N5ESMR) level 1 data have been reprocessed to estimate global sea ice concentration from 11 December 1972 to 16 May 1977. The full data set is available in the CEDA Archive: DOI:10.5285/8978580336864f6d8282656d58771b32 at a grid resolution of 25 x 25 km$^2$ and a daily timestep (Tellefsen et al. (2025a)). A new methodology using locally and sea-
5 sonally variable algorithm coefficients called tie points has been used to calculate the sea ice concentration in both first-year ice and multi-year ice in the Arctic and the radiometrically distinct ice types A and B in Antarctica. Validation of sea ice concentration using Arctic sea ice charts from the US National Ice Center shows an overestimation of open-water SIC of up to 20 % in some places and an underestimation of SIC in sea ice covered regions near the ice edge. Validation also shows that local dynamical tie points (LDTP) improve the mapping of sea ice concentration for different types of ice, while estimates
of the extent of sea ice are identical to the previous processing of the same data in Kolbe et al. (2024). A new set of quality control (QC) filters has been developed that discards far fewer data points (57.7 % reduction) than the filters in the previous processing. The data set therefore closes significant gaps in the sea ice concentration and sea ice extent record compared to the earlier data record. Of the 1.6 billion data points recorded by the satellite, 23.0 % have been discarded. 1136 days during the 1616-day period from 1972 to 1977 are covered (at least partly), which gives estimates of the mean monthly sea ice minimum
and maximum extent in the Arctic and Antarctica during this period, except for some dropouts in 1973 and 1975.

## 1   Introduction

The extent of sea ice is an important indicator of climate change because sea ice affects and responds to changes in the energy balance of the Earth's surface and the ocean and atmospheric circulation (Notz and Stroeve (2016); Masson-Delmotte et al. (2021); Silvano et al. (2025); Moore et al. (2022)). Long observational records of the extent of sea ice, dating back to the
20 beginning of the 1970s, are critical to understanding these processes and to validate models (Fogt et al. (2022); Goosse et al. (2024)). For example, air-sea heat flux in the Nordic Seas, linked to atmospheric and ocean circulation, and sea ice, experienced a persistent change in the late 1970s (Moore et al. (2022)) and in the mid-1970s the extent of Antarctic sea ice experienced a drop, which was comparable to and yet less persistent than the more recent drop in 2016 (Goosse et al. (2024)).





Satellite microwave radiometer data have provided a detailed record of these recent global changes in sea ice extent (SIE) and concentration (SIC) from 1978 to today (see, e.g., Tonboe et al. (2016); Lavergne et al. (2019); Parkinson (2019)). However, recently, experimental satellite microwave radiometer data from the Nimbus satellite program have been used to extend these sea ice extent records further back to the 1970s (Parkinson et al. (1987); Kolbe et al. (2024, 2025); Tellefsen et al. (2025b)). The data set described here is part of that effort.

The methodology developed for processing the data set in Tonboe et al. (2016) to reduce noise and achieve stability of the sea ice climate data record (CDR) when processing data from nine different multichannel instruments (Nimbus-7 SMMR, 6 DMSP SSM / I and 2 DMSP SSMIS) is also applicable to processing historical and experimental data sets from Nimbus-5 and 6 ESMR with poor calibration (Kolbe et al. (2024); Tellefsen et al. (2025b)). The same methodology has also been applied to process microwave radiometer sounder data from the Nimbus-6 SCAMS instrument not traditionally used for sea ice mapping (Kolbe et al. (2025)) when NASA made these data sets available at level 1 in 2016 (GSFC (2016)).

This methodology for processing these data sets first reduces the regional variability of $T_B$ due to geophysical noise sources (wind-induced roughness of the ocean surface, atmospheric temperature, water vapor, cloud liquid water and ice temperature) using atmospheric reanalysis data and a radiative transfer model (RTM). Secondly, the SIC algorithm is calibrated to the actual ice and open water $T_B$ signatures measured by the instrument to avoid biases from instrument drift and calibration, inter-sensor differences, reanalysis data, RTM and seasonal / inter-annual variations in the ice and water $T_B$ signatures. Third, residual uncertainties are quantified using an uncertainty model (Tonboe et al. (2016)). We follow those steps here.

In 2023, the ESA Sea Ice Climate Change Initiative (ESA CCI) released the *Nimbus-5 ESMR Sea Ice Concentration, version 1.0* (v1.0) dataset (Tonboe et al., 2023), using the methodology of Tonboe et al. (2016) and processed as described in Kolbe et al. (2024). However, because Nimbus-5 ESMR (N5ESMR) is a one-channel instrument, there is an ambiguity between ice type and sea ice concentration in the v1.0 data set. Furthermore, data suffer from drift in housekeeping data, which led the data quality control (QC) filters in Kolbe et al. (2024) to discard about half of the data. Especially the last two years of instrument operation from 1975-1977 is sparsely covered after QC. These two issues have been addressed in this new *Nimbus-5 ESMR Sea Ice Concentration, version 1.1* (v1.1) Tonboe et al. (2025) processing of the N5ESMR data, by: 1) applying regional tie points and taking into account the radiometric differences between the ice types, and 2) applying QC filters that only discarded individual erroneous data points and were adapted to calibration variations.

An independent evaluation of the v1.1 dataset has been conducted by Kern (2025) comparing it with Landsat-1 high-resolution optical imagery in 1974, which showed that the statistical performance parameters are as good as for the evaluation of modern SIC records (Kern et al. (2022)).

The objectives of this study are to reprocess the N5ESMR data to estimate the extent and concentration of sea ice for both polar regions, including the uncertainties of the estimates.





## 2   The Nimbus-5 ESMR instrument and data

The Nimbus-5 Electrically Scanning Microwave Radiometer (ESMR) was a single channel (horizontally polarized 19.35 GHz) Dicke microwave radiometer mounted in front of the satellite. The 83 x 83 cm phased array antenna controlled the 78 scan positions across the track, giving varying incidence angles and spatial resolution from approximately 25 km at nadir to approximately 150 km at $63^o$ at the edges of the swath symmetrically around nadir. The orbit altitude of Nimbus-5 was $\sim 1100$ km and the width of the swath on the Earth's surface was $\sim 3100$ km covering both poles in 13.4 daily orbits. A combined side-lobe and incidence angle correction has been applied to the Level 1 data provided online at (GSFC (2016)). The angle of incidence correction, which we were unable to remove separately from the side lobe correction, means that brightness temperatures ($T_B$) at horizontal polarization do not decrease as a function of the angle of incidence as expected for ocean and sea ice. Level 1 data have been co-located with atmospheric and surface parameters from ERA5 global atmospheric reanalysis simulations from the European Centre for Medium-Range Weather Forecasts (ECMWF) so that every $T_B$ measurement is associated with a simulated estimate of atmospheric water vapor, total column water, air surface temperature, surface wind speed, and sea ice concentration (Bell et al. (2021)). Subsequently, these data are used together with a radiative transfer model to reduce the geophysical noise in the $T_B$ data (Kolbe et al. (2024)). Because this processing step may introduce biases from the ERA5 data and the RTM the tie points are therefore updated after the correction to reduce these potential biases.

### 2.1   The National Ice Center sea ice charts for comparison

The National Ice Center sea ice charts in digital format is used for comparison with the Nimbus-5 ESMR v1.1 SIC. During the period when Nimbus-5 ESMR was operated, weekly ice charts were based on satellite data (Nimbus-5 ESMR, NOAA 2, 3, 4 VHRR visual and infrared imagery) combined with aerial reconnaissance data and observations from ships (Fetterer (2016)). The different sources of information are combined in a manual analysis in the ice chart (Partington et al. (2003)). Uncertainties in the sea ice concentration according to Partington et al. (2003) of $\pm 5$ % to $\pm 10$ %, arise from poor resolution of printed satellite imagery and cloud cover that obscures the surface when using VHRR data (Fetterer (2016)). The sea ice concentrations of the ice chart are generally higher than the sea ice concentrations derived from satellite microwave radiometer data (Partington et al. (2003)).

## 3   Methodology

The processing of v1.1 is a direct modification of the methodology to that of the v1.0 product. The methodology of Kolbe et al. (2024) is therefore largely followed in this paper, with the exception of the choice of quality filters (Sect. 3.1), and with an additional post processing step utilizing local dynamical tie points (LDTP) to correct the SIC predictions (Sect. 3.2).





### 3.1 QC filtering

The raw N5ESMR data set contains erroneous measurements, as a result of instrumental flaws and calibration failure. For v1.0, Kolbe et al. (2024) used various QC filters specifically for each error type, and we also follow this approach here. Although they were successful in removing most erroneous data points, the filters also removed a significant amount of valid data, especially during the latter half of the N5ESMR operation after 10 November 1974. This has been found to be the result of a systematic change in the instrument noise level, reflected in the housekeeping data used for filtering. This and other filters have been revisited here and we have therefore created a new QC filtering scheme that relies solely on features present in the $T_B$ data and not in the housekeeping data. The new scheme has been tested on all available swaths, and the threshold values are selected by qualitative inspection of data point distributions and outliers.

For the new scheme, four types of filters have been constructed that each target specific faulty patterns in the data. These are; i) value filters where thresholding is used to sort data within physically valid $T_B$ intervals, ii) pixel filters which evaluate the validity of a pixel based on its neighbors, iii) sweep filters that remove one or more sequential sweeps, and iv) swath filters that detect if the entire swath is corrupt.

#### 3.1.1 Value filter

The value filter works by simply removing pixels with values outside the range specified in Equation (1).

$$90K < T_B(i,j) < 310K \tag{1}$$

Since $T_B$ of sea ice does not exceed 273.15 K, we could have limited the range to this, as done in Kolbe et al. (2024). However, to preserve the full distribution including noise and poor calibration of $T_B$s within our area of interest, we have chosen an upper threshold of 310 K.

#### 3.1.2 Pixel filter

The pixel filter removes values significantly different from the local median value as shown in Eq. (2):

$$|T_B(i,j) - \text{med}_{3\times3}[T_B(i,j)]| < 75K \tag{2}$$

Where $\text{med}_{3\times3}[T_B(i,j)]$ is the median of a local neighborhood of 3 by 3 pixels. The result of this filter is reminiscent of the value filter in Eq. (1), even though the pixel filter detects local errors that are still within the valid $T_B$ range (see Eq. (1)).

#### 3.1.3 Sweep filters

The sweep filters focus on the relative difference between two consecutive sweeps, as described in Equation (3):

$$\Delta T_B(i) = \text{med}_j \left( \frac{T_B(i,j) - T_B(i+1,j)}{T_B(i,j)} \right) \tag{3}$$

$\Delta T_B(i)$ is used in three different ways to remove specific patterns in the dataset;




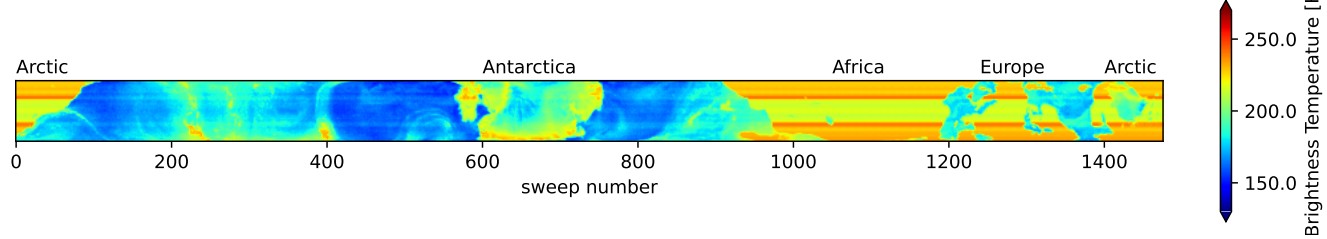

**Figure 1.** Example of the types of swaths the swath filter is meant to detect. In this swath, $T_B$ over land and ice is constant along track, as the true $T_B$ is above 220 K.

1. If $|\Delta T_B(i)| > 0.09$, sweep $i$ and $i+1$ are removed.

2. If $|\Delta T_B(i)| > 0.09$ and the sweep $i$ is within 25 rows from the start or end of the swath, all pixels in between are removed.

3. If $|\Delta T_B(i)| > 0.06, |\Delta T_{B(i+k)}| > 0.06, k \leq 25$ and $\text{sign}(\Delta T_B(i)) = -\text{sign}(\Delta T_B(i+k))$, all rows between the two are removed.

The sweep filter removes most "zones" where consecutive errors occur in entire sweeps. This was implemented as the swaths often contain sections where consecutive sweeps have a significantly different value compared to their surroundings.

We also discarded sweeps where more than 25 % of the data points are missing from other sweeps both before and after the current sweep (in a range of 25 sweeps). This resolves issues where errors are affected by multiple filters, as the effects of individual filters might otherwise counteract one another.

### 3.1.4 Swath filter

In the Nimbus-5 data Catalogues vol. 2. (GSFC, 1972-1974) it is stated that:

"Starting at 11:10 GMT during orbit 1062 (28 February [1973]) and continuing to the end of this catalog period, a malfunction reduced the ESMR instrument response to the range between 110 K and 220 K of brightness temperature. Before the malfunction, the range was between 110 K and about 300 K."

For the remainder of the operation, this error appeared periodically for varying time intervals ranging from singular orbits to several months. An example of this error is shown in Fig. 1. The "stripy" pattern in the figure is the result of sidelobe and incidence angle corrections done by NASA, which warp the 220 K saturation limit to different values for each data point position across the swath. $T_B$ for FYI are typically between 230 K and 240 K, which is above the saturation limit.

Although some information, e.g. detection of ocean/ land/ ice can still be derived from the corrupted swaths like the one in Fig. 1, the saturated $T_B$ observations cannot be used for SIC with the methods presented in this paper. Therefore, a filtering scheme inspired by Kolbe et al. (2024) has been applied which removes swaths with repeating values in 5 consecutive sweeps.





For some corrupt swaths, repeats occur only in every other sweep, and therefore the filter has been designed to compensate for this: The filter is only activated when the erroneous pattern is detected more than 100 times in a single swath. The filter is presented in Equation (4).

$$\sum_{k=0}^{5} D(i+k) = 0, \qquad \sum_{k=0}^{5} D(i+2k) = 0, \qquad D(i) = T_B(i,j) - T_B(i+1,j) \tag{4}$$

## 3.2 The Local Dynamical Tie Point (LDTP) approach

Kolbe et al. (2024) uses a one-channel SIC algorithm, based on dynamical tie points for ice ($T_{p,ice}$) and water ($T_{p,water}$) to derive $SIC$ shown in Eq. (5), where $i$ and $j$ represent the spatial coordinates, $t$ is time.

$$SIC(i,j,t) = \frac{T_B(i,j,t) - T_{p,water}(t)}{T_{p,ice}(i,j,t) - T_{p,water}(t)} \tag{5}$$

The tie points $T_{p,ice}$ and $T_{p,water}$ were the mean $T_B$ in the areas covered by the respective surface types in the ERA5 climatology and estimates of the sea ice concentration from the data itself. The 15-day running mean tie points were calculated to reduce day-to-day noise.

The problem with this approach is that it does not account for variations in $T_B$ for different types of sea ice. The different ice surfaces show different spectral properties and spatial and temporal dependence of $T_{p,ice}(i,j,t)$. At 19.35 GHz, there is a significant difference between first-year ice (FYI) and multi-year ice (MYI), with the latter having a lower emissivity due to increased scattering from the sea ice microstructure (Tonboe (2010)). For v1.0, this resulted in an SIC bias, where MYI SIC is underestimated.

The $T_{p,ice}(i,j,t)$ is described as a function of two tie points $T_{p,FYI}(t)$ and $T_{p,MYI}(t)$, together with a ratio parameter $c_{MYI}(i,j,t)$, as shown in Eq. (6):

$$T_{p,ice}(i,j,t) = c_{MYI}(i,j,t)\, T_{p,MYI}(t) + (1 - c_{MYI}(i,j,t))\, T_{p,FYI}(t) \tag{6}$$

This model has two parameters, $c_{MYI}(i,j,t)$ and $SIC(i,j,t)$ to predict, but there is only one observable variable ($T_B$). Modern microwave radiometers (e.g. SMMR, AMSR2) measure more frequencies and polarizations; and, for these, there are algorithms that utilize this multimodality to resolve the ambiguity (Comiso et al. (1997)). Here we store the spatio-temporal information, which can itself be regarded as a distinct modality to resolve the SIC and ice type ambiguity. This is the logic behind the LDTP algorithm.

### 3.2.1 Algorithm structure

A flow chart illustrating the fully implemented algorithm is shown in Fig. 2. This implementation uses the regridded $T_B$ from v1.0, where the RTM $T_B$ corrections have already been applied. The algorithm is based on the assumption that $T_{p,ice}(i,j,t)$ is directly observable locally when $SIC$ is constrained to 100 %:

$$T_{p,ice}(i,j,t) \approx T_B(i,j,t) \quad \leftrightarrow \quad SIC(i,j,t) \approx 100\,\% \tag{7}$$



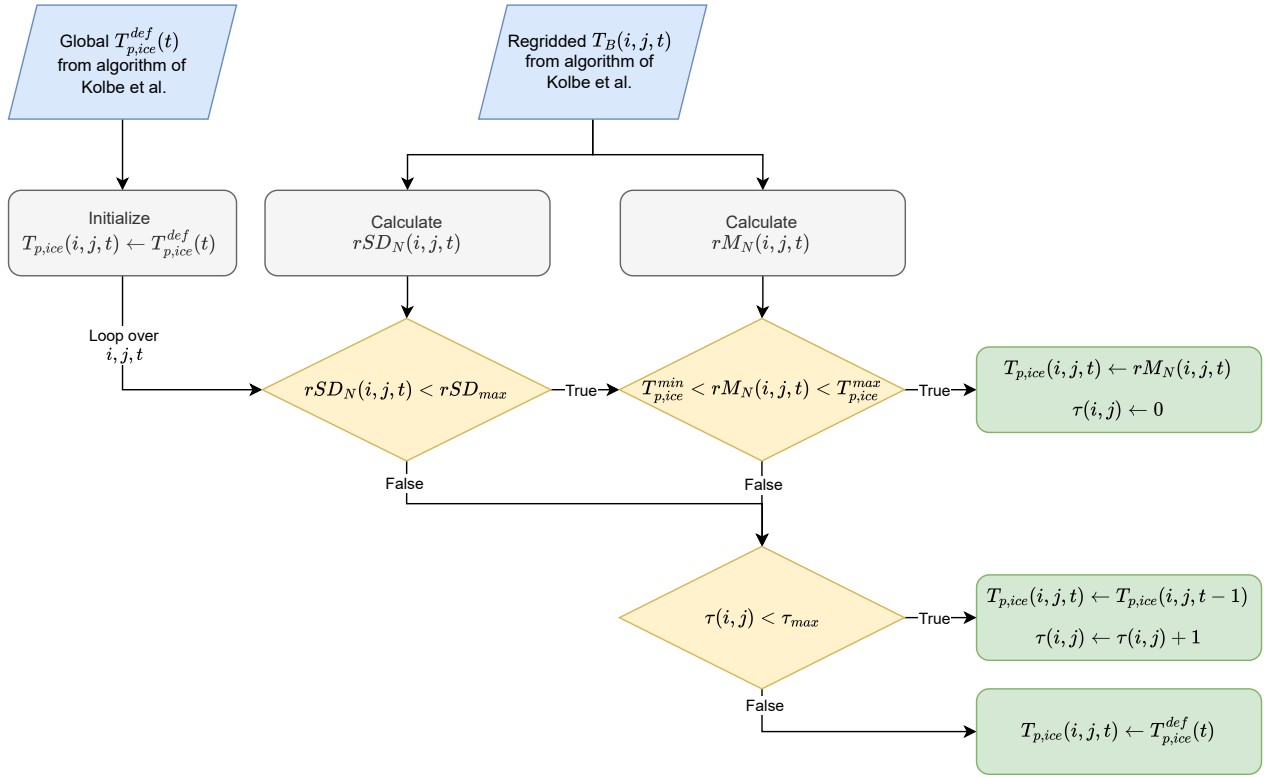

**Figure 2.** Flow chart for the LDTP algorithm. For the final data product, we have reprocessed the algorithm in reverse time order once to re-initialize $T_{p,ice}(i,j,t)$ for the first couple of days of satellite operation.

Assuming that $T_{p,ice}(i,j,t)$ is stable in time, this can be used as a way to find ice tie points locally even when $SIC(i,j,t) \leq 100\,\%$. Thus, a measure of when $SIC(i,j,t)$ is $100\%$ is needed. For this, $T_B(i,j,t)$ is assumed to have a low variance over time when the SIC is capped (0 % or 100 %), compared to the intermediate SIC where the $T_B$ variance is high. This is explained by concentration fluctuations, as ice drift and melt / refreezing cause intermediate SIC, and thus $T_B$, to rarely remain stable over multiple days.

A measure of stability is therefore needed to select the local tie point. We have used the running standard deviation ($rSD$ - see eq. (8)) to measure $T_B$ stability. This depends on the running mean $rM$, which is calculated as shown in eq. (9).

$$rSD_N(i,j,t) = \frac{1}{N-1} \sum_{n=-N/2}^{N/2} \sqrt{(rM_T(i,j,t) - T_B(i,j,t+n))^2} \tag{8}$$

$$rM_N(i,j,t) = \frac{1}{N-1} \sum_{n=-N/2}^{N/2} T_B(i,j,t+n) \tag{9}$$





There are two parameters to consider for thresholding based on $rSD$; the period length to evaluate over, $N$ and the thresholding value itself $rSD_{max}$. The choice of these two parameters is discussed in Section 3.2.2.

$T_B(i,j,t)$ could also be stable when $SIC(i,j,t) = 0\,\%$. To address this, a threshold value $T_{p,ice}^{min}$ is selected such that
$rM_N(i,j,t) > T_{p,ice}^{min}$ for $SIC(i,j,t) = 100\,\%$. To also avoid erroneous tie points during the summer melt, a maximum tie point value, $T_{p,ice}^{max}$, is chosen. With these criteria, the evaluation of cases where $SIC(i,j,t) = 100\,\%$ is:

$$SIC(i,j,t) = 100\,\% \quad \leftarrow \quad rSD_N(i,j,t) < rSD_{max} \quad \wedge \quad T_{p,ice}^{min} < rM_N(i,j,t) < T_{p,ice}^{max} \tag{10}$$

By combining Eq. (7) and Eq. (10), $T_{p,ice}(i,j,t)$ is chosen and $SIC(i,j,t)$ is derived using Eq. (5). It is assumed that $T_{p,ocean}$ is independent of space $(i,j)$ and the tie points of Kolbe et al. (2024) are therefore reused for open water.

The LDTP approach cannot be applied everywhere since not all areas experience $SIC(i,j,t) = 100\,\%$. In regions (e.g. the ocean, East Greenland current and along the ice edge) where a stable signature cannot be established, we therefore use the hemispheric tie points from Kolbe et al. (2024).

Furthermore, some areas are only rarely covered by ice and because $T_{p,ice}$ change over time, we have implemented a maximum age limit of a local tie point, $\tau_{max}$, of 180 days. If a local tie point is older than 180 days, then the hemispheric tie
point from Kolbe et al. (2024) is chosen. The tie point age limit was chosen to be as short as possible while still being longer than the longest periods of missing data and the melt season.

Finally, the algorithm needs an initialization period to determine which tie points to pick, and therefore we have run the algorithm forward and backward once for all dates and have picked the resulting tie points as initial values for the final evaluation; thus, we only use the tie points of Kolbe et al. (2024) for the first initialization or if $\tau_{max}$ is reached.

### 3.2.2 Parameter tuning

To select the parameters $rSD_{max}$, $N$, $T_{p,ice}^{min}$, and $T_{p,ice}^{max}$, we analyzed areas representing FYI and MYI during the period 01 January 1973 - 01 March 1973. The manually selected areas are shown in Fig. 3 by investigating the $T_B$ patterns. We avoided latitudes above 88°N, as the data here are derived entirely from scan positions with an incidence angle greater than 50.1° (from where incidence angles increase by more than 1° per scan position).

We found that of a total of 142544 observations in time and space, only 35 observations had $T_B$ values outside the interval $[205\,K, 255\,K]$ (16 below and 19 above); therefore, we chose $T_{p,ice}^{min} \leftarrow 205\,K$ and $T_{p,ice}^{max} \leftarrow 255\,K$.

For selecting $rSD_{max}$ and $N$, we have analyzed the mean $rSD$ distribution for all grid points from the selected reference areas and $N$. The result is shown in Fig. 4. On the one hand, low $rSD_{max}$ makes $T_{p,ice}$ updates too infrequent, and $T_B$ noise could mimic changes in SIC. On the other hand, too high $rSD_{max}$ results in $T_{p,ice}$ being updated in cases where $SIC$ is
changing, thus biasing its prediction. Furthermore, the distribution of $rSD$ is affected by the duration of the period, $N$, as seen in Fig. 4. Low values of $N$ increase the likelihood of a biased update of the tie point. Therefore, $N$ should ideally be long. However, a too long $N$ results in infrequent tie point updates and delayed response to changes in the ice $T_B$s.

Based on Fig. 4, we decided to select a period length of $N = 15$ and decrease it to a minimum of $N = 7$ in case of data gaps. Data gaps were frequent after 14 September 1975, since Nimbus-5 ESMR data were only received every two days to





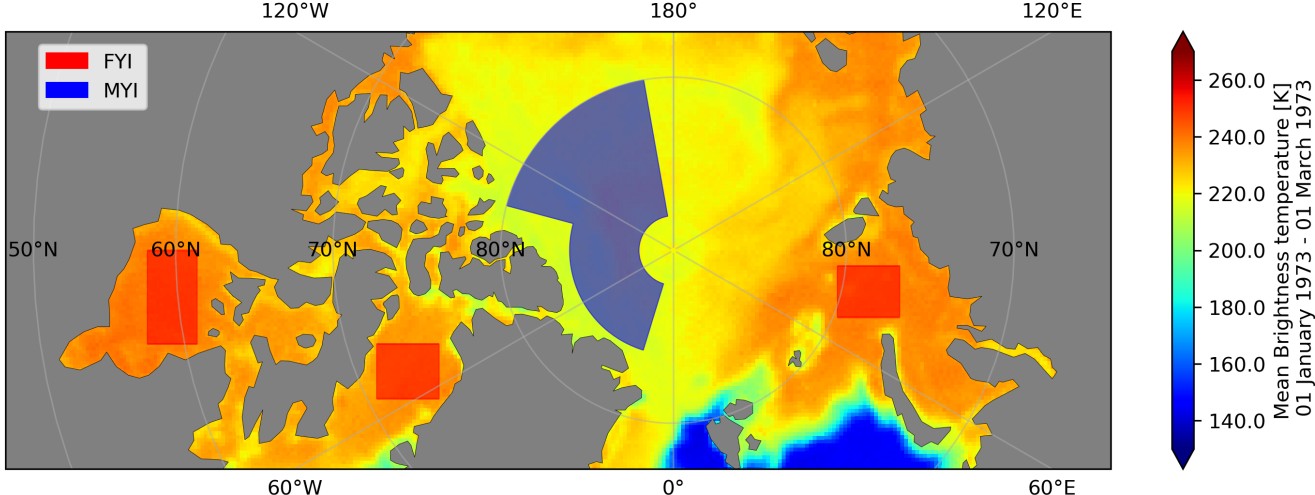

**Figure 3.** Areas selected as reference for parameter tuning of the algorithm. Areas containing both FYI and MYI have been selected to ensure that the algorithm is tuned to both ice types.

prioritize resources for Nimbus-6 (Parkinson et al., 1999). $rSD_{max}$ is set to 3.737, which is the mean value $rSD$ for $N = 15$ in all reference areas. The mean does not change significantly between $N = 7$ and $N = 15$. Fig. 4 shows that MYI has a mean $rSD$ slightly lower than this (and vice versa for FYI), which means that the algorithm accepts more MYI tie points than FYI tie points, which, contrary to Kolbe et al. (2024), means that the algorithm performs better on MYI than FYI.

## 4 Results and discussion

This section presents the effects of the new filtering scheme, and we describe the differences between the LDTP algorithm and the v1.0 algorithm. In addition, the SIC of the new and old algorithms are compared with the SIC from the National Ice Center (NIC) ice charts, and we have derived new global SIE estimates, which we compare to those derived from v1.0.

### 4.1 The data QC filters

An example of the effect of all QC filters (excluding the swath filter) individually and together is shown along with the results 215 of the old filtering scheme from Kolbe et al. (2024) in Fig. 5.

When comparing "All" and "Old" in Figure 5, it is seen that significantly fewer valid data points are removed with the new scheme. However, some "miscalibration" errors still occur, as seen around sweep 200 for "All", but they are less prominent, and the new filters do detect some errors not detected by the old filters, e.g. near sweep 250.

In general, the old QC filters removed significantly more than the new QC filters, especially after 10 November 1974, as 220 seen in Table 1. The temporal pattern for data removal is further examined in Figure 6, which compares the fraction of data



**Figure 4.** Boxplot chart showing mean $rSD$ distribution for the reference areas of Fig. 3 for the period 01 January 1973 - 01 March 1973. Bars illustrate how many tie point updates occur on average for each cell given each period length $N$.

removed for all swaths using the old and new filters. The figure clearly shows a pattern change around 10 November 1974 for the old QC filters, which is not seen in the new QC filters.

A peak in errors is observed around January 1977, which we found to be the result of a periodic calibration error where the mean signal changes approximately every twentieth sweep; thus, the new filter detects this error, as intended.

### 4.2 LDTP algorithm outputs

Fig. 7 illustrates the tie point and $T_B$ differences between the Kolbe et al. (2024) and the LDTP algorithms for the Baffin Bay area of Fig. 3.

Baffin Bay cycles between open water during summer and consolidated FYI during winter, as shown in Fig. 7. The Kolbe et al. (2024) ice tie points are consistently lower than the tie points of the LDTP algorithms, and the LDTP tie points follow



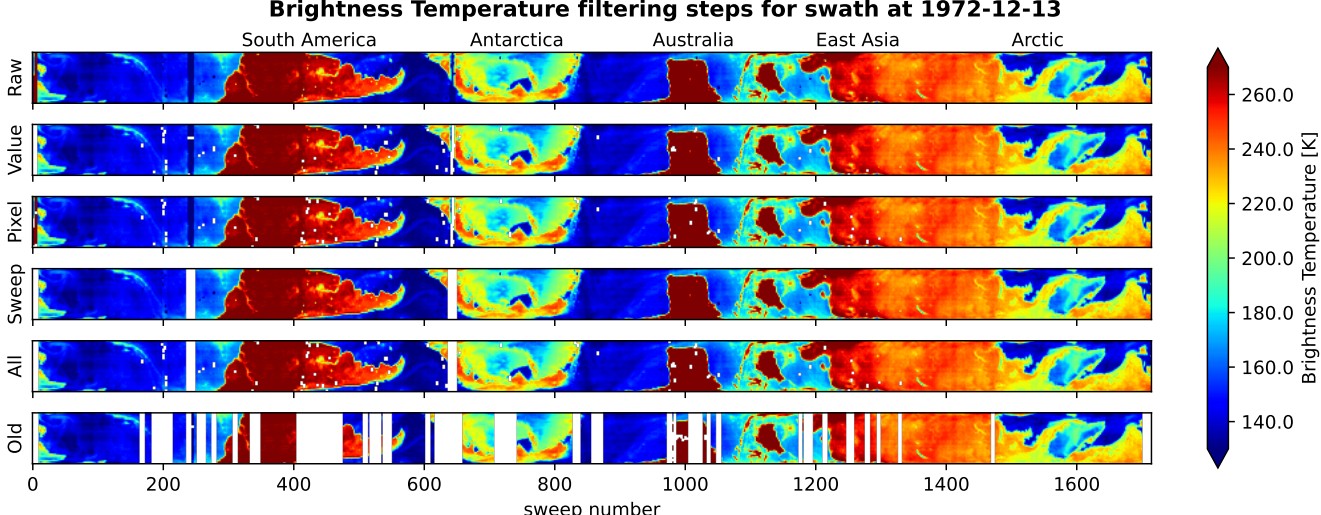

**Figure 5.** Display of new filters applied individually and together on a select swath (from 13 December 1972), along with the result from applying the old scheme to the same swath.

**Table 1.** Fraction of data removed by the old and the new filtering scheme, both overall, and constrained to before and after 10 November 1974.

| Data removal | Overall | Excluding swaths completely removed | | |
|---|---|---|---|---|
| | | Overall | Before 10 November 1974 | After 10 November 1974 |
| Old Filters | 54.4 % | 40.0 % | 2.3 % | 85.3 % |
| New Filters | 23.0 % | 1.5 % | 0.7 % | 2.4 % |

the $T_B$ closely. However, the LDTP tie points do follow the Kolbe et al. (2024) tie points closely by the end of 1973, 1976, and 1977, due to a mixture of long melt season and missing data, which means the tie point age limit is activated and the old, hemispheric, tie points are used. Figure (7) shows that the new tie points in Baffin Bay replicate the $T_B$ variability.

Figure 8 shows the spatial performance of $SIC$, $T_B$, local tie points, $SIC$ standard error, $rSD$ and cell updates for both the Arctic and Antarctic near the maximum extent of the ice. We see that the ice tie points for the Arctic are distinct for FYI on

the Siberian shelves, Baffin Bay and Hudson Bay, and MYI in the central Arctic. The ocean has a uniform signature, which is explained by the fact that the hemispheric Kolbe et al. (2024) open water tie point is selected. MYI is no longer distinguishable on the $SIC$ map as in Kolbe et al. (2024). $rSD$, is a stability measure with low values near 3 in the central Arctic, smaller regions in the Ross Sea and Weddell Sea, and high values near the ice edge. Updates appear to be more frequent in the Arctic than in Antarctica, which could be due to the fact that consolidated MYI has a lower $rSD$ than FYI (see Section 3.2.2) and

that there is little MYI in Antarctica compared to the Arctic.




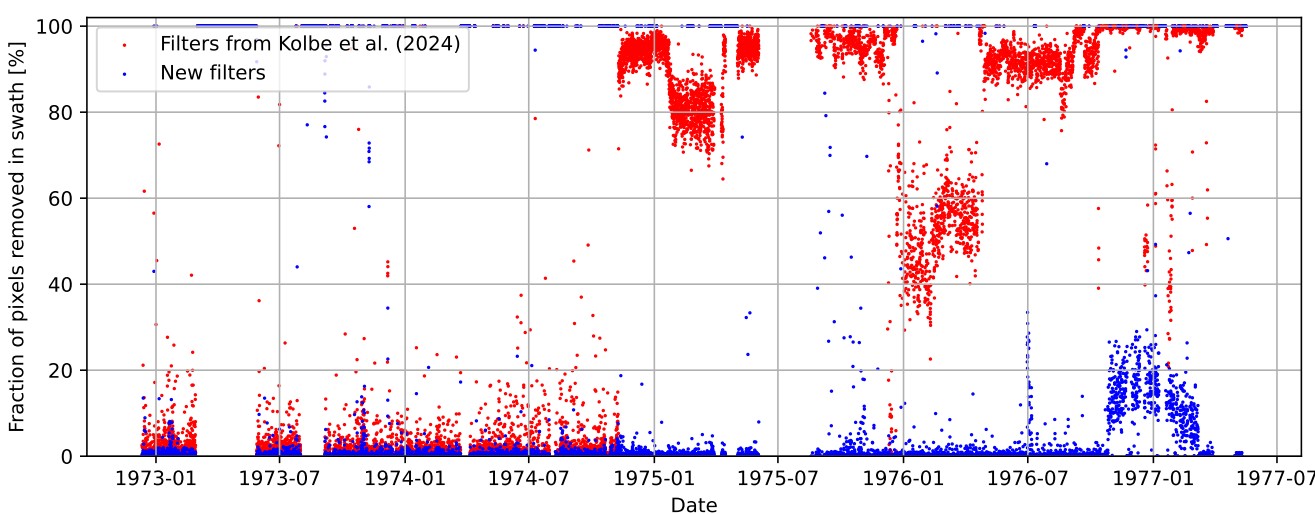

**Figure 6.** Amount of data being removed given the old vs. new filtering scheme. The figure shows that the old scheme in general removes more data than the new, especially following 1974-11.

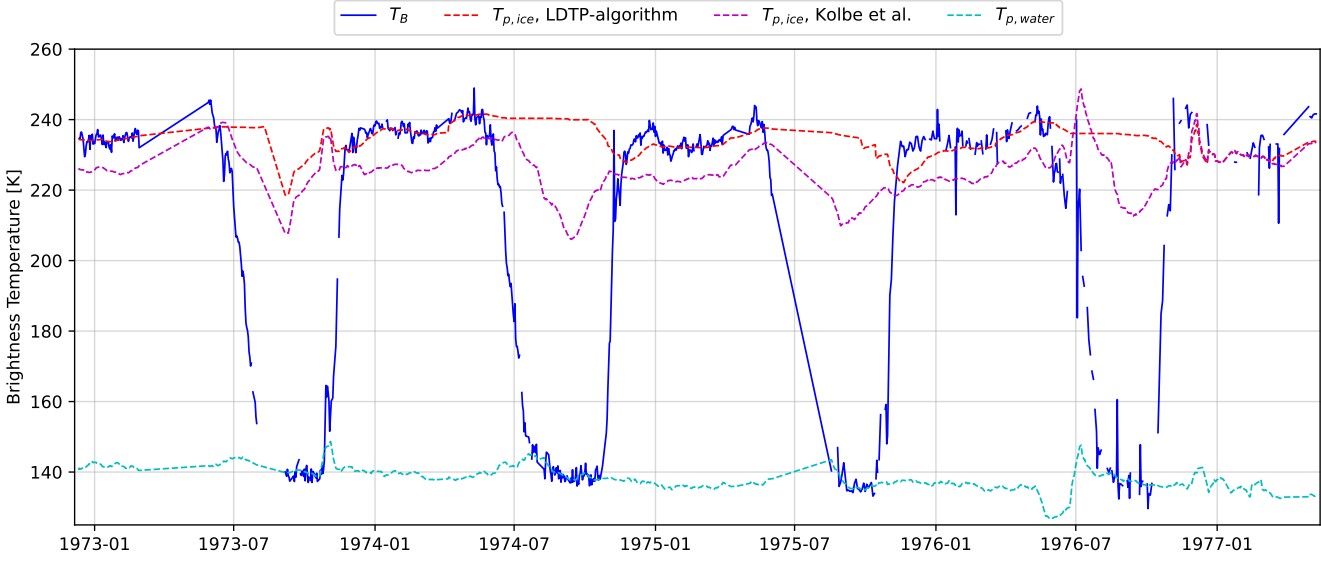

**Figure 7.** Average brightness temperature and tie points for the Baffin Bay subset (same as seen in Figure 3). Ice tie points from both Kolbe et al. (2024) algorithm and the LDTP algorithm are shown for comparison. It is clear that the LDTP algorithm follows the ice signature of 100 % $SIC$ more closely than the old algorithm.





**Figure 8.** SIC, $T_B$, LDTP-based Ice tie points, $SIC$ Standard Error, $rSD$ and updated tie points for the Arctic (01 February 1974) and the Antarctic (01 September 1974), created using the LDTP algorithm. $rSD$ is the running standard deviation for a 15 day period centered at the current date, while "tie points updated" are cells which tie point values have been updated for the present data, as $rSD$ has gone below the set threshold of 3.737 K.



### 4.3 Comparison with other SIC products

The new daily v1.1 SIC (Tonboe et al., 2023) are compared with v1.0, and Arctic ice charts from the National Ice Center (NIC) (Fetterer (2016)) in Fig. 9. We have used the zones of Fig. 3, to evaluate the biases over different surface types between the different products.

The $SIC_{v1.1} - SIC_{v1.0}$ difference plot in Fig. 9 shows that the LDTP algorithm correctly has near 100 % SIC over MYI with an average difference of 9. 0 %, and an average difference over FYI of -2.0 %. The positive $SIC_{v1.1} - SIC_{v1.0}$ difference on Arctic MYI is a manifestation of the overestimated MYI tie point in Kolbe et al. (2024) and vice versa for FYI. This issue has been resolved in the new LDTP processing, as illustrated by the difference $SIC_{v1.1} - SIC_{NIC}$ in Fig. 9.

     In the Arctic $SIC_{v1.1}$ and $SIC_{NIC}$ comparison, the differences are related to the spatial resolution of ESMR and naviga-
tional ice charts in the Laptev Sea and along the ice edge in general. Over the central Arctic Ocean pack ice, the LDTP SIC is near 100 % as expected, while the ice chart is assigned a constant SIC of 95 % as an indication that the real SIC is between 90 % and 100 % (Partington et al. (2003)). Keeping this 5 % bias in mind, we find an average $SIC_{v1.1} - SIC_{NIC}$ bias over MYI of 2.5 % and 2.9 % for FYI, meaning that $SIC_{v1.1}$ on average predicts that the SIC is 2. 7 % higher than the 95 % SIC from the digitized ice chart. Ice charts are not available for Antarctica.

The $T_B$ of thin ice / new ice is between ice and open water $T_B$ and $SIC$ (Ivanova et al. (2015); Tonboe and Toudal (2005)) and the negative $SIC_{v1.1} - SIC_{NIC}$ difference in the Greenland Sea is probably caused by the presence of new ice Wadhams and Wilkinson (1999); Comiso et al. (2001); Tonboe and Toudal (2005). The ice charts from the 1970's do not have information on ice type (except a mapping of fast ice), but an inspection of the ice charts after 1995 when ice type information became available in the charts shows that the Greenland Sea and the Barents Sea have large fractions of new ice/ thin ice during mid-
winter. In open water, the LDTP SIC is overestimated compared to the ice chart due to geophysical noise that affects the SIC of the microwave radiometer (Tonboe et al. (2022)).

### 4.4 Sea ice extent estimates

The SIE derived for both hemispheres using v1.0 and v1.1 is shown in Fig. 10. Tabular versions of the new data are found in Appendix B. The extent is calculated as the total area with an SIC greater than 30 %, and only months with more than 99 %
coverage are used.

     As expected, the two schemes are very similar, as the LDTP algorithm of v1.1 mainly affects multi-year ice SIC, which is not detected at SIC < 30 %. The main difference between algorithms for computing SIE is the temporal coverage, which is improved by the new approach as a result of the updated QC filtering scheme. Some small differences can be seen in the latter half of 1974, likely a result of summer melt, which violates the assumptions behind both algorithms, that is, that a stable
signature for 100 % SIC (the tie point) can be estimated. The summer period sea ice area and SIC estimates should therefore be used with caution (see Sect. 4.5).

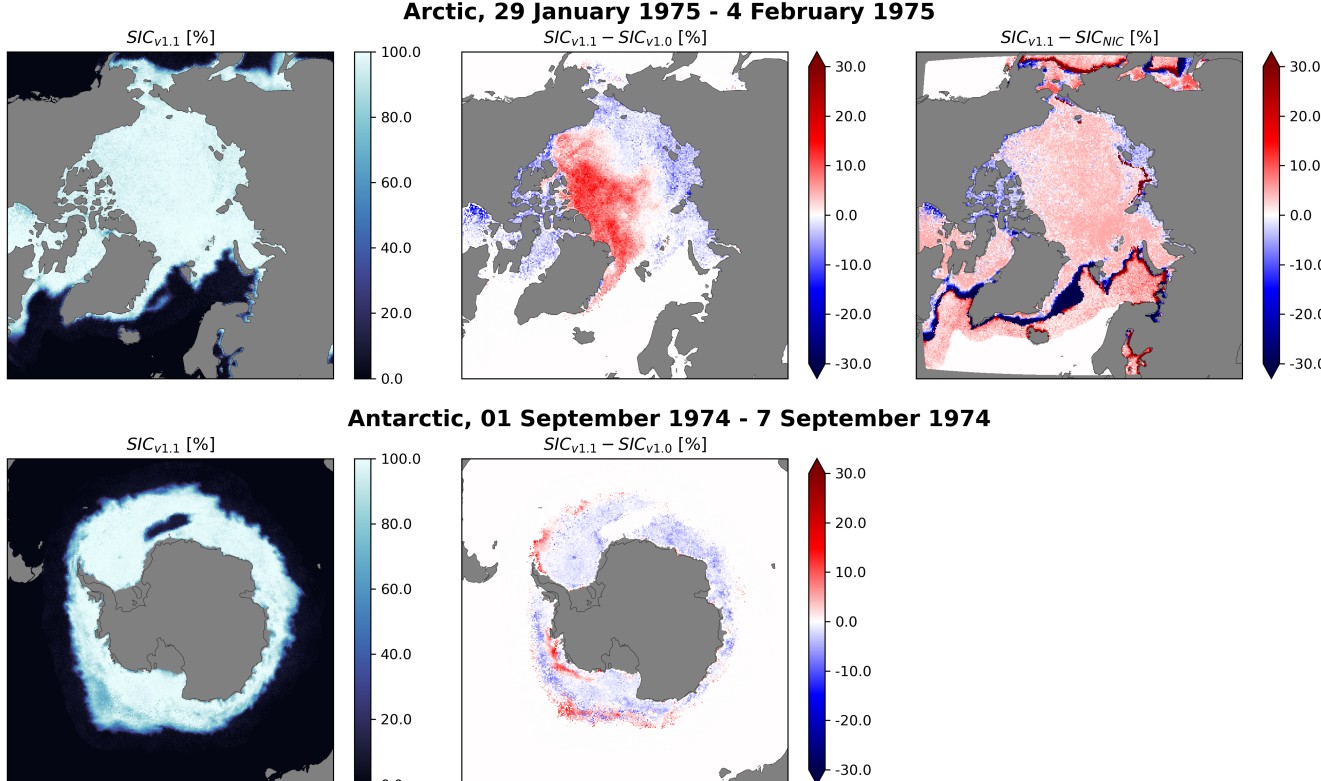

**Figure 9.** The top row shows Northern Hemisphere sea ice concentrations (SIC) of v1.1 (left), the v1.1 SIC - v1.0 SIC difference (middle) and the v1.1 SIC - NIC ice chart SIC difference (right), for the week covered by the ice chart 29. Jan. -4 Feb. 1975. An example of Southern Hemisphere SIC is depicted in the bottom row with the v1.1 SIC (left) and the v1.1 SIC - v1.0 SIC difference (middle) for the week 1. Sep. - 7. Sep. 1974.

## 4.5 The summer melt season sea ice concentration

It is clear that the requirement for the tie points, namely that the tie point signature is that of 100 % SIC, is very difficult to achieve in summer when cracks and leads remain open and melt ponds form at the surface. This issue remains even in modern sea ice concentration data sets such as those of Lavergne et al. (2019). During summer melt in the Arctic, melt ponds start to form on the surface of the ice, and at the same time the emissivity of the snow and ice in between the melt ponds changes. Eventually, late in the melt season, some of the melt ponds will melt through the floe and come into direct contact with the ocean below. However, because of the very shallow penetration depth ($\delta_p$) of microwaves in water ($\delta_p < 1$ mm @ 19 GHz, Ulaby et al. (1986)), the melt ponds will have a signature similar to that of the water in the cracks, and the leads in-between the floes and the SIC mapped with microwave radiometers are a measure of the ice surface fraction (Kern et al. (2016)). In Antarctica, melt ponds are rare, but there are still melt-induced emissivity changes of snow and ice (Istomina et al. (2022)).

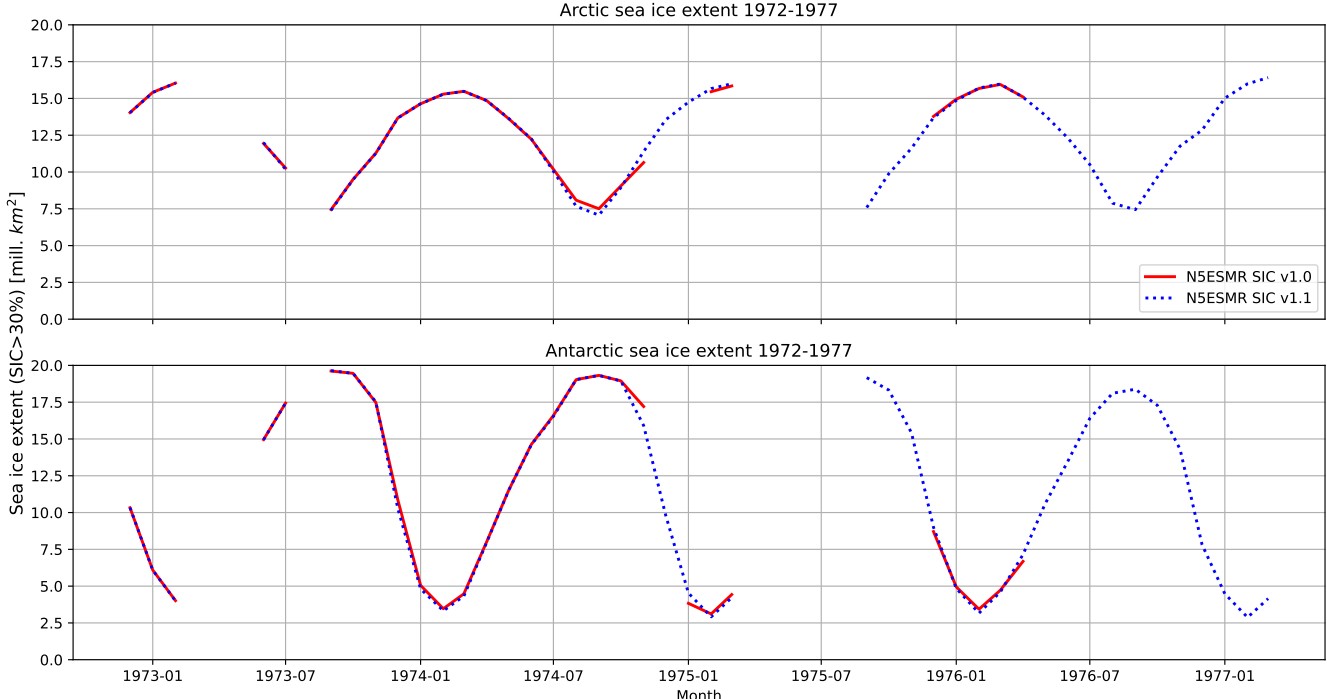

**Figure 10.** Monthly SIE for the Arctic and the Arctic calculated with a threshold of 30 % SIC. The v1.1 product generally preserve more data compared to v1.0. Some dropouts are still observed in 1973 and 1975, and are the result of the saturation error described in sect. 3.1.

Although SIC estimates have high uncertainties during the summer, the algorithm still recovers quickly when the melt ends. Melt-induced changes in emissivity affect the SIC estimates and the sea ice area, while the SIE estimates are mainly affected by the level of open water noise (Andersen et al. (2007)).

## 5 Dataset

The v1.1 dataset is publicly available (Tonboe et al., 2025). It is gridded on a 25 km EASE 2.0 grid (Brodzik et al., 2012) and is available as daily NetCDF files for all dates where valid N5ESMR data are available. To maintain consistency with v1.0 and other similar SIC datasets, we have maintained file structure, variable names, and status flags, which means that the dataset can be used as a direct replacement for v1.0. An overview of the data set variables can be found in the Tab. A1. There is a description of the status flags in Tab. A2. We have not included the LDTP algorithm specific variables: $rSD$, $rM$, $T_{p,ice}$, $\tau$ and $T_{p,water}$ to be consistent with the v1.0 data structure. However, these parameters can be derived using the available code (see below) and the corrected brightness temperatures (TB_corr) of the v1.1 dataset.





# 6 Data availability

The *ESA Sea Ice Climate Change Initiative (Sea_Ice_cci): Nimbus-5 ESMR Sea Ice Concentration, version 1.1* dataset gener-
295 ated from the methods presented here, is available in the CEDA Archive. DOI:10.5285/8978580336864f6d8282656d58771b32
(Tonboe et al., 2025). The same dataset, along with an updated v1.0 product where only QC filters are corrected can be found
at DTU data. DOI: 10.11583/DTU.27835929.v2 (Tellefsen et al., 2024).

# 7 Code availability

All data processing code is available on GitHub (https://github.com/EHTellefsen/N5ESMR-SIC-processing). The repository
includes the old and new QC filtering scheme, scripts to derive the v1.0 product with the old and new QC filtering scheme,
code to upgrade v1.0 to v1.1, and script to create the plots of this paper (data not included).

# 8 Conclusions

Two issues, sea ice type and SIC ambiguity and data QC in Kolbe et al. (2024) have been addressed in this new version 1.1
processing of the N5ESMR SIC data: 1) A new methodology has been developed that uses locally and seasonally variable
algorithm tie points to process the N5ESMR data to retrieve the sea ice concentration in both first-year ice and multi-year ice
in the Arctic and the radiometrically distinct ice types A and B in Antarctica. The new processing has resolved the SIC and
the ice-type ambiguity in v1.0. An independent evaluation of the N5ESMR SIC dataset in Kern (2025) using high-resolution
optical Landsat-1 imagery as a reference shows that the bias, standard deviation and correlation coefficient is comparable to
a similar evaluation of modern SIC CDR's (Kern et al. (2022)). In addition, a comparison with Arctic sea ice charts from the
310 US National Ice Center shows a bias of up to 20 % in open water near the ice edge, which is higher than the noise in modern
multichannel radiometer records (Ivanova et al. (2015). However, using a SIC threshold of 30 %, the extent of sea ice is close
to identical to the v1.0 processing in Kolbe et al. (2024) and avoids open water noise to affect the estimates of SIE.

2) New QC filters have been developed that use 77 % of the N5ESMR data points. This is a significant improvement
compared to the previous processing in which only 45 % of the data points were used. This improvement in coverage therefore
closes significant gaps in the sea ice concentration and sea ice extent record which gives estimates of the mean monthly Arctic
sea ice minimum in 1974, **1975**, **1976** and the maximum extent in 1974, 1975, 1976, 1977 and the Antarctica minimum in
1974, 1975, 1976, **1977** and the maximum in 1973, 1974, **1975**, **1976**. The years in bold are additions since Kolbe et al.
(2024). The maximum and minimum mean monthly gaps from June 1975 to March 1976 can be closed to some extent with the
SCAMS microwave radiometer data onboard the NIMBUS 6 satellite (Kolbe et al. (2025)). The next phase of the European
Space Agency's Climate Change Initiative project will focus on the combination of satellite SIC records from different sensors
in the 1970's for a more complete assessment of the global SIC and SIE.





## Appendix A: Dataset parameters

**Table A1.** Description of the variables for the v1.1 dataset.

| | |
|---|---|
| ice_conc | Sea ice concentration corrected for land spillover and filtered using ERA5 2 m air temperature and climatological masks [%]. |
| raw_ice_conc_values | Raw sea ice concentration estimates as retrieved by the algorithm [%] |
| total_standard_error | Total uncertainty (one standard deviation) of sea ice concentration [%] |
| smearing_standard_error | Smearing uncertainty of sea ice concentration [%] |
| algorithm_standard_error | Algorithm uncertainty of sea ice concentration [%] |
| status_flag | Status flag bits for the sea ice concentration, as described in Table A2 |
| Tb_corr | Corrected brightness temperatures [K] |
| Tb | Uncorrected brightness temperatures [K] |
| time | Time of data [year, month, day] |
| xc | x coordinate of projection [km] |
| xy | y coordinate of projection [km] |
| lat | Latitude [°] |
| lon | Longitude [°] |

**Table A2.** Description of the status flags for the v1.1 dataset.

| | |
|---|---|
| no flag/flag 0 | Nominal retrieval by the SIC algorithm |
| flag 1 | Land |
| flag 2 | Lake |
| flag 4 | SIC is set to zero by the open water filter |
| flag 8 | SIC value is changed to correct for land spillover effects |
| flag 16 | The 2 m air temperature flag is raised at this position |
| flag 32 | Coast |
| flag 64 | SIC is set to zero since the position is outside the maximum sea ice climatology |
| flag 128 | Point is not accepted but no other flags are raised |



## Appendix B: Sea ice extent

**Table B1.** Monthly v1.1 Arctic sea ice extent (Area with SIC > 30 %) [mill.km$^2$]

| Year | Jan | Feb | Mar | Apr | May | Jun | Jul | Aug | Sep | Oct | Nov | Dec |
|------|------|------|------|------|------|------|------|------|------|------|------|------|
| 1972 |      |      |      |      |      |      |      |      |      |      |      | 14.03 |
| 1973 | 15.40 | 16.04 |      |      |      | 11.99 | 10.19 |      | 7.40 | 9.49 | 11.27 | 13.67 |
| 1974 | 14.64 | 15.29 | 15.48 | 14.85 | 13.65 | 12.25 | 10.06 | 7.67 | 7.07 | 8.95 | 11.39 | 13.55 |
| 1975 | 14.75 | 15.67 | 16.00 |      | 13.73 |      |      |      | 7.59 | 9.88 | 11.62 | 13.69 |
| 1976 | 14.87 | 15.71 | 15.98 | 15.08 | 13.86 | 12.31 | 10.51 | 7.88 | 7.44 | 9.65 | 11.76 | 12.85 |
| 1977 | 15.02 | 16.00 | 16.41 |      |      |      |      |      |      |      |      |      |

**Table B2.** Monthly v1.1 Antarctic sea ice extent (Area with SIC > 30 %) [mill.km$^2$]

| Year | Jan | Feb | Mar | Apr | May | Jun | Jul | Aug | Sep | Oct | Nov | Dec |
|------|------|------|------|------|------|------|------|------|------|------|------|------|
| 1972 |      |      |      |      |      |      |      |      |      |      |      | 10.33 |
| 1973 | 6.08 | 4.00 |      |      |      | 14.95 | 17.44 |      | 19.64 | 19.46 | 17.52 | 10.27 |
| 1974 | 4.82 | 3.32 | 4.33 | 7.97 | 11.50 | 14.62 | 16.53 | 19.04 | 19.32 | 18.94 | 15.91 | 9.82 |
| 1975 | 4.53 | 2.91 | 4.22 |      | 10.52 |      |      |      | 19.17 | 18.32 | 15.42 | 9.00 |
| 1976 | 4.84 | 3.17 | 4.60 | 7.15 | 10.58 | 13.48 | 16.40 | 18.10 | 18.38 | 17.29 | 14.30 | 7.82 |
| 1977 | 4.48 | 2.88 | 4.15 |      |      |      |      |      |      |      |      |      |



*Author contributions.* EHT has developed software for data processing and analysis and wrote the draft manuscript, RTT has developed the
325 prototype processing software and supervised the project, and WMK has developed software and conducted the validation. All authors have
contributed to the writing and end editing of the manuscript.

*Competing interests.* None of the authors has any competing interests.

*Acknowledgements.* This study was supported by the European Space Agency's Climate Change Initiative (ESA/PB-EO(2021)9) under the
European Earth Watch Programme. We would like to thank NASA for making the Nimbus-5 ESMR level 1 data available online (GSFC
(2016)), the Copernicus Climate Change Service for making the ERA5 data available (Copernicus-Climate-Change-Service (2023)) and the
National Snow and Ice Data Center and the United States National Ice Center for the ice charts (U.S. National Ice Center (2006)).





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
