# Peer review of "Mapping sea ice concentration using Nimbus-5 ESMR and local dynamical tie points"

_Earth System Science Data, 2025_

## Referee Comment (RC1)

**Comment on essd-2025-660:**

The paper presents a new sea ice concentration dataset derived from reprocessed data from the Nimbus-5 ESMR radiometer. The new processing addresses two main issues that characterized the previous version of the dataset (Kolbe et al., 2024). First, the authors incorporate locally and seasonally defined dynamical tie points to account for differences in the signatures of ice types, including improved estimation of multi-year ice. Second, they develop new quality control filters that allow to discard fewer observation points compared to the previous methodology. This represents a real advancement over the previous study and the dataset provides added value by closing significant gaps compared to the earlier processing. This enabled a more complete dataset at early times (late 1972 to 1977).

The dataset is straightforward to use both in QGIS and using python scripts for the Northern and the Southern hemisphere. The manuscript is generally clearly structured and well written. I think that a few minor revisions could further clarify the achievement and validation of the new dataset.

**General comments:**

- While the algorithm tuning and dataset description for the Arctic regions are clear and well-elaborated, I believe the description of the dataset and the application of the algorithm to Antarctica is somewhat rushed. I understand the RTM model accounts for initial radiative differences between the two regions, but the manuscript would benefit from more explicit acknowledgment of these hemispheric distinctions throughout. This suggestion primarily concerns rephrasing and making explicit statements where regional differences exist. For instance, when it comes to one of the paper's primary novelties (local dynamical tie points) it would be important to clarify the differences in the tuning and the validation processes more explicitly throughout the manuscript. To summarise, I believe a more thorough discussion of the hemispheric differences in the quality of the dataset would strengthen the paper. I will elaborate on this concern in the following points:

    - The absence of an equivalent for Fig. 3 for Antarctica suggests that a clearer statement about the choice of the areas selected as references and the subsequent tuning for the Arctic and their transferability for Antarctica would be necessary.
    - Lines 53-54 state that the objective is to evaluate uncertainties and estimates for both polar regions; therefore, a consistent comparison of dataset quality throughout the paper is necessary for both hemispheres.
    - Equation (6) is presented only for FYI-MYI, but the same equation applies to ice types A and B. This should be explicitly specified in the text. Additionally, while FYI-MYI and ice types A and B are distinguished in terms of ice and snow electromagnetic properties, after their introduction, ice types A and B are not mentioned again until the conclusion.

- Line 211 presents the validation with ice charts (Arctic region) and then the use of the word 'global' for the SIE estimates.
- Section 4.3 contains minimal discussion of Antarctica. Similarly, the Antarctic subplots in Figure 9 appear to lack corresponding commentary in the text. (The same applies to the few comments on Fig. 8 for the Southern hemisphere). Perhaps adding a dedicated discussion paragraph on the differences in results and uncertainties in the dataset between the polar regions would enhance clarity.

- Section 4.5 appears somewhat disconnected from the preceding results discussion. I recommend either strengthening the connection to the dataset presented in this study or incorporating this content into another section where the link to the data would be clearer.

**Technical comments:**
- Line 8: 'in some places' , could it be rephrased in a more accurate way?
- line 67: I understand it is mentioned in the Introduction and cited with Kolbe et al. 2024 but I would specify again the name of the radiative transfer model (RTM).
- Line 85: remove 'and' .
- Line 88: make 'this' more explicit.
- Line 92: grammatically correct but 'that each' makes the sentence difficult to read, I would rephrase and avoid using the semicolon before the list .
- Line 163 to 166, I would add at least one reference for this statement.
- Lines 206 to 208: the different standard deviation behavior for FYI and MYI, along with the subsequent interpretation, could be better addressed in the Results and Discussion section rather than in the tuning description, given that this represents an interpretation and comparison with the previous algorithm rather than a methodological detail.
- Figure 3: I suggest to label the area selected given that you are referring to them later (for instance, Baffin Bay).
- Figure 7: it is good to see how the LDTP tie points follow TB more closely for the sea ice, however it is not straightforward to understand the dip of the ice tie points when the ocean tie point should be assumed. This is just a suggestion on the visualization choice.
- Figure 8: The use of the same color scale for different TB ranges across the subplots may reduce clarity. Please consider whether individual color scales for each subplot would more effectively highlight the results shown in the figure.
- Line 239-240: as mentioned, I think ice types A and B should not be used interchangeably with FYI and MYI, as the signatures are not necessarily the same. Tie points update: it is great to see the method has improved compared to previous work in the Arctic, where tie points are mostly sensitive to ice age. Is it straightforward to attribute the fewer tie point updates in Antarctica to MYI-FYI differentiation in this region?
- Line 265: specify 'coverage'.
- Line 278: @ → at
- Figure 10 caption: correct Arctic to Antarctic.
- Line 304: change to full stop if you write with capital A after 1)

- Line 307: the sentence describing ice-type ambiguity resolution could be rephrased for improved clarity.
- Line 307-310 : while this independent validation provides valuable support for the analysis quality, the authors should clarify what strengthens confidence in the Arctic product (evaluation from Kern (2025) and ice charts) and what validates the Antarctic product as well (Kern et al. 2022).
- Reference to subplots throughout the manuscript would be easier for the reader with the addition of panel letters to multi-panel figures.

---

## Author Response (AR1)

**Reviewer 1**

*Reply: Thanks for the constructive review and your comments. We have provided a point-by-point reply below and we have updated the MS accordingly. Your suggestions have made the MS much clearer and more complete. In particular, we have added more discussion of Antarctic aspects which we feel has made the MS much more balanced between the two hemispheres.*

Comment on essd-2025-660: The paper presents a new sea ice concentration dataset derived from reprocessed data from the Nimbus-5 ESMR radiometer. The new processing addresses two main issues that characterized the previous version of the dataset (Kolbe et al., 2024). First, the authors incorporate locally and seasonally defined dynamical tie points to account for differences in the signatures of ice types, including improved estimation of multi-year ice. Second, they develop new quality control filters that allow to discard fewer observation points compared to the previous methodology. This represents a real advancement over the previous study and the dataset provides added value by closing significant gaps compared to the earlier processing. This enabled a more complete dataset at early times (late 1972 to 1977). The dataset is straightforward to use both in QGIS and using python scripts for the Northern and the Southern hemisphere. The manuscript is generally clearly structured and well written. I think that a few minor revisions could further clarify the achievement and validation of the new dataset. General comments:

● While the algorithm tuning and dataset description for the Arctic regions are clear and well-elaborated, I believe the description of the dataset and the application of the algorithm to Antarctica is somewhat rushed. I understand the RTM model accounts for initial radiative differences between the two regions, but the manuscript would benefit from more explicit acknowledgment of these hemispheric distinctions throughout. This suggestion primarily concerns rephrasing and making explicit statements where regional differences exist. For instance, when it comes to one of the paper's primary novelties (local dynamical tie points) it would be important to clarify the differences in the tuning and the validation processes more explicitly throughout the manuscript. To summarise, I believe a more thorough discussion of the hemispheric differences in the quality of the dataset would strengthen the paper. I will elaborate on this concern in the following points:

The absence of an equivalent for Fig. 3 for Antarctica suggests that a clearer statement about the choice of the areas selected as references and the subsequent tuning for the Arctic and their transferability for Antarctica would be necessary.

*Reply: The most apparent problem with the version 1.0 dataset was the underestimation of SIC over multiyear ice and at the same time overestimation over first-year ice in the Arctic. The latter was less visible because of the truncation at SIC>100% but it resulted in suppressed first-year ice SIC variability. So we started addressing this Arctic problem, but it is clear that the SIC estimates are benefitting from the LDTP approach over the radiometrically distinct ice types in Antarctica as well. The regions in Antarctica where SIC in v1.0 was underestimated compared with v1.1 were relatively local (along the ice edge in the Weddell and Ross Seas and in the central Amundsen Sea). However, the SIC in the rest of*

*Antarctica was overestimated in v1.0 compared to v1.1, meaning an underestimation of the SIC variability in v1.0. This is a serious issue because even small changes in the near 100% SIC have a large impact on fluxes. So while the v1.0 SIC/ ice type issue is more apparent in the Arctic, there are large benefits of the v1.1 SIC in Antarctica as well. We have included these arguments in the discussion.*

Lines 53-54 state that the objective is to evaluate uncertainties and estimates for both polar regions; therefore, a consistent comparison of dataset quality throughout the paper is necessary for both hemispheres.

*Reply: The methodology is consistent for both hemispheres and at the same time both hemispheres have specific tie-points (LDTP) which are tuned to the radiometrically distinct ice types - this is a requirement for unbiased SIC's. Everything else is the same for both hemispheres to make SIC and SIC uncertainty estimates comparable and consistent.*

*For example, we have chosen not to change the thresholds on variability (rSD) for updating tie-points between the two hemispheres or the way noise reduction is done or land-spill-over correction to give consistency in the methodology between the hemispheres.*

*However, the Antarctic ice cover is in many places more dynamic than in the Arctic in terms of ice drift and changes in the emissivity and this variability is reflected in SIC uncertainties which are in general higher in Antarctica. We believe that the higher SIC uncertainty estimates in Antarctica compared to the Arctic are because of sea ice environment differences between the Arctic and the Antarctic.*

Equation (6) is presented only for FYI-MYI, but the same equation applies to ice types A and B. This should be explicitly specified in the text. Additionally, while FYI-MYI and ice types A and B are distinguished in terms of ice and snow electromagnetic properties , after their introduction, ice types A and B are not mentioned again until the conclusion.

*Reply: Thanks for pointing this out. We have named the radiometrically distinct types FYI and MYI in both the Arctic and in Antarctica and included an explanation that radiometric distinction is not only due to ice age. Especially in Antarctica but also in the Arctic, the radiometric distinction is also a function of snow cover.*

- Line 211 presents the validation with ice charts (Arctic region) and then the use of the word 'global' for the SIE estimates.

*Reply: We have deleted this sentence.*

- Section 4.3 contains minimal discussion of Antarctica. Similarly, the Antarctic subplots in Figure 9 appear to lack corresponding commentary in the text. (The same applies to the few comments on Fig. 8 for the Southern hemisphere). Perhaps adding a dedicated discussion paragraph on the differences in results and uncertainties in the dataset between the polar regions would enhance clarity.

*Reply: We have added a discussion about the Antarctic tie point update frequency and rSD in section 4.2, and we have discussed patterns in the Antarctic tie point choices and their effect on estimated SIC in v1.1 and also compared to the v1.0 SIC in section 4.3.*

● Section 4.5 appears somewhat disconnected from the preceding results discussion. I recommend either strengthening the connection to the dataset presented in this study or incorporating this content into another section where the link to the data would be clearer.

*Reply: Section 4.5 has been integrated with 4.3*

Technical comments:

Line 8: 'in some places' , could it be rephrased in a more accurate way?

*Reply: Thanks, we have added specific sites "…where open ocean meets land and near the ice edge and an underestimation of SIC in sea ice covered regions near the ice edge especially in the Greenland Sea."*

line 67: I understand it is mentioned in the Introduction and cited with Kolbe et al. 2024 but I would specify again the name of the radiative transfer model (RTM).

*Reply: We have added which domains: "…the radiative transfer model (RTM) for the ocean, ice and atmosphere…"*

Line 85: remove 'and' .

*Reply: Thanks, we have deleted "and".*

Line 88: make 'this' more explicit.

*Reply: We have specified "this": "Although they were successful in removing most erroneous data points, the filters also removed a significant amount of valid data, especially during the latter half of the N5ESMR operation after 10 November 1974. The removal of valid data-points has been found to be the result of a systematic change in the instrument noise level, reflected in the housekeeping data used for filtering."*

Line 92: grammatically correct but 'that each' makes the sentence difficult to read, I would rephrase and avoid using the semicolon before the list.

*Reply: Thanks, we have rephrased the sentence.*

Line 163 to 166, I would add at least one reference for this statement.

*Reply: Thanks. This is pretty central to the use of LDTP and we have added an explanation and two references (Andersen et al. 2007 and Kwok, 2002) to support the assumptions.*

Lines 206 to 208: the different standard deviation behavior for FYI and MYI, along with the subsequent interpretation, could be better addressed in the Results and Discussion section rather than in the tuning description, given that this represents an interpretation and comparison with the previous algorithm rather than a methodological detail.

*Reply: The last part of the paragraph has been moved to discussion as suggested.*

Figure 3: I suggest to label the area selected given that you are referring to them later (for instance, Baffin Bay).

*Reply: All four areas have been labeled with a letter (a,b,c,d) and a name (Hudson Bay, Baffin Bay…)*

Figure 7: it is good to see how the LDTP tie points follow TB more closely for the sea ice, however it is not straightforward to understand the dip of the ice tie points when the ocean tie point should be assumed. This is just a suggestion on the visualization choice.

*Reply: Thanks for pointing this out. We have changed the illustration and corresponding text to reflect a single pixel instead of the mean trends over an area to more clearly reflect the cause and effect of the algorithm's choice of tie points.*

Figure 8: The use of the same color scale for different TB ranges across the subplots may reduce clarity. Please consider whether individual color scales for each subplot would more effectively highlight the results shown in the figure.

*Reply: We have changed the color scale of the ice tie point plots (fig. 8C and 8I), to make it distinct from their brightness temperature counterparts.*

Line 239-240: as mentioned, I think ice types A and B should not be used interchangeably with FYI and MYI, as the signatures are not necessarily the same. Tie points update: it is great to see the method has improved compared to previous work in the Arctic, where tie points are mostly sensitive to ice age. Is it straightforward to attribute the fewer tie point updates in Antarctica to MYI-FYI differentiation in this region?

*Reply: Thanks, we have clarified in the text that the radiometrically distinct types (both in the Arctic and in Antarctica) are called FYI and MYI and that the differences are due to both ice age and snow type and potentially other things.*

Line 265: specify 'coverage'.

*Reply: We define total coverage as all ocean pixels in the 25 km EASE 2.0 grid within the climatological mask, i.e. the area of interest. This has also been specified in the text.*

Line 278: @ → at Figure 10 caption: correct Arctic to Antarctic.

*Reply: Thanks.*

Line 304: change to full stop if you write with capital A after 1) –

*Reply: Thanks.*

Line 307: the sentence describing ice-type ambiguity resolution could be rephrased for improved clarity. - -

*Reply: Thanks for pointing this out. The sentences have been rephrased and references to the figure have been added.*

Line 307-310 : while this independent validation provides valuable support for the analysis quality, the authors should clarify what strengthens confidence in the Arctic product

(evaluation from Kern (2025) and ice charts) and what validates the Antarctic product as well (Kern et al. 2022).

*Reply: We do not have ice charts covering Antarctica and there has not been an independent study comparing the Antarctic SIC with Landsat or other high resolution optical data similar to Kern (2025). The Kern (2025) validation was almost exclusively over Arctic first-year ice and the validation shows that the v1.1 SIC is virtually unbiased. We think that this is also an indication that the v1.1 SIC in Antarctica is less biased than in v1.0. This explanation is also included in the text.*

Reference to subplots throughout the manuscript would be easier for the reader with the addition of panel letters to multi-panel figures.

*Reply: We have added reference letters (A, B,...) for fig. 4, 8, 9 and 10, and referenced accordingly in the text.*

**Reviewer 2**

Review of "Mapping sea ice concentration using Nimbus-5 ESMR and local dynamical tie points" by Tellefson et al.

*Reply: Thanks for the constructive review and your comments. We have provided a point-by-point reply below and we have updated the MS accordingly. Your suggestions have made the MS much clearer and more complete.*

Summary

This paper presents an improved method to derive sea ice concentrations from the Nimbus-5 ESMR sensor using local dynamical tie points and improved quality control measures. The improvements in quality control allow considerably more viable data than the previous method yielding a more complete time series. The local tie points reduce biases in concentration and provide a higher quality product.

General Comment

This is an incremental but valuable improvement to the previous ESMR sea ice concentration product. Though only a single-channel instrument and some issues with quality, ESMR is a valuable resource because it can extend the passive microwave record by about 7 years. The previous version of the product was very helpful to make the ESMR data useful for sea ice studies. This version makes substantial improvements, yielding more complete data and less biased fields. The methodology is sensible and explained clearly. It is good to see that the data and software are only available. The manuscript is acceptable with only minor revisions in response to the comments below.

Specific Comments (by line number):

13-15: This sentence is a bit unwieldy and confusing. It talks about some dropouts in 1973 and 1975 – I presume these are monthly averages. In the same sentence, it discusses

minimum and maximum extents – I assume here these are the lowest annual months (Mar and Sep in the Arctic). I assume the dropouts in 1973 and 1975 refer to months in general, not min/max months, but it isn't clear. A rewrite – maybe separating out the mean months and the mention of min/max months into two sentences.

*Reply: This paragraph has been rewritten now specifying the missing Antarctic minimum and Arctic maximum (which is, with the v1.1 filters, only March 1973).*

17-18: There are many, many papers that address the importance of sea ice extent as an indicator for climate change. Listing only four is fine, but I would include "e.g.". Also, the order of the citations seems random. Not sure what the standards are for this journal, but usually they are listed in temporal order.

*Reply: Thanks. We have added e.g and ordered them chronologically.*

19: It is not only "Long observational records", but "Long and consistent" records that are key. Without consistency, the observational record is not going to be useful.

*Reply: Thanks. We have added the word "consistent".*

23: "a more recent drop" instead of "the more recent drop" – using "the" presumes a familiarity with the Antarctic record that a reader may not have.

*Reply: True. We have included a reference (Turner et al. 2017) and a short description of the 2016 Antarctic sea ice retreat.*

27: "from an earlier Nimbus satellite program instrument" – since SMMR on Nimbus-7 has long been a part of the standard PM record, you can't just say "data from the Nimbus satellite program". Actually, I think you could rework this paragraph and the next to be a bit clearer. In the next paragraph (29-24), you discuss the PM sensors – SMMR, SSMI, SSMIS. Perhaps mention that first and then say, "Recently, experimental satellite microwave data from the Nimbus-5 and Nimbus-6 program have been used…"

*Reply: Thanks. This section has been reorganized, presenting the NIMBUS program first and then explaining about the specific sensors and datasets.*

57: I think many readers may not be familiar with a "Dicke microwave radiometer". I think just "passive microwave radiometer" may be sufficient, but otherwise, "Dicke" should be explained.

*Reply: The radiometer type is important for the radiometric resolution and the number of calibration points needed. However, since this is not important for the discussion we have deleted "Dicke" as you suggested.*

58-59: It might be worth noting that other PM radiometers in the standard time series (SMMR, SSMI, SSMIS) are conically-scanning. So, their incidence angle and IFOV are constant, in contrast to EMSR.

*Reply: Thanks, we have added an explanation of the across track scanning N5ESMR and the conically scanning radiometers: N6ESMR and those after that you mention.*

72: The NIC charts used Nimbus-5 ESMR, so the comparison with this product are not completely independent. I think the comparison is useful because (I believe) the ESMR data used in the NIC charts were manually analyzed (maybe just looking at TBs), so the analysis methods are independent. But I think it is good to emphasize the lack of complete independence and explain why the comparison is still useful - either here or in the results section where the comparisons are made.

*Reply: Thanks for the suggestion. We have clarified that in the data section.*

100-101: Why was 310 K chosen? As you note, ice won't have a TB > 273.15 K. I can see some buffer to account for instrument noise, but 310 K seems rather arbitrary. Why not 300 K? Or even 280 K?

*Reply: Good question. During the lifetime of ESMR we noted that shifts in the mean brightness temperatures could exceed 20 K (June 1976, https://doi.org/10.5194/essd-16-1247-2024, Fig. 2.) related to temporary instrument calibration issues and in addition we were not sure of the absolute calibration so then we selected a solid margin above 273.15 K. In fact, we have selected the upper range of the radiometer dynamic range. In practice, it does not matter if the threshold is 310 K or 320 K because the erroneous data detected by this threshold are anyway much higher (and other filters will detect those errors) and we keep all valid data anywhere on Earth with a threshold of 310K. This has been clarified in the text as well.*

147-148: "where MYI SIC is underestimated". Do you mean "where SIC is underestimated in MYI regions"? In other words, I think you mean total SIC is underestimated due to the presence of significant MYI? I guess the total SIC is underestimated because the MYI is underestimated, but I think the point here is that total SIC is biased low because the MYI TB signature is not accounted for.

*Reply: Yes, thanks, we have changed it as suggested.*

149: This paragraph is discussing the new method, in contrast to the old method discussed in the previous paragraph, correct? I think a transition phrase would be helpful, "In our new method, the Tp,ice…" Also, add "value" after "Tp,ice(i,j,t)".

*Reply: Yes, thanks, we have changed it as suggested.*

Figure 3: Though the figure is pretty clear, a couple suggestions:

(1) the MYI in the map seems to be a different color than the legend – the blue in the MYI legend actually looks more like blue shade of open water in the map.

(2) the contrast of the FYI red with the underlying TB color scale could make it difficult for some people to distinguish between the two (the red isn't too far off from the dark orange). I can make things out, so maybe not a big deal, but changing the color scales – either of the FYI/MYI boxes or the TB values – would be clearer. For example, you could use a viridis (or similar) scale for TB and then the red/blue for the FYI/MYI would stand out clearly. Or just change the FYI/MYI colors to be more

distinct from the rainbow/jet TB scale. You could also consider colored box outlines instead of solid boxes to mark the FYI/MYI areas.

*Reply: We have decided to keep "jet" as colormap for the TB scale to stay consistent with the remaining figures of the paper. Instead, we changed the colors of area-of-interest boxes to be more distinguishable.*

Figure 4: Are the "whiskers" the min and max values? And the box represents one standard deviation around the mean rSD. I guess it is a bit confusing because it is a plot of statistics around an SD value. And often a boxplot is used to denote median and quartiles, though I assume that is not the case here because it states "mean rSD".

*Reply: Thanks for raising this. Lines 210-212, Figure 4, and the figure text has been updated to clarify how the plot was generated. The boxplots do represent standard quartile distribution, but were made on the per pixel mean rStd's which were generated by first calculating the mean rStd across time for each pixel given N.*

*One can argue statistics should be based on the raw rSD values. In practice this would widen the distribution, but keeping the medians roughly static. We argue for meaning across time first, as the LDTP algorithm operates on each pixel on its own, and per pixel updates are more important than the total global frequency of tie point updates.*

*As part of the clarification, it became apparent that we referenced the wrong value for $rSD_{max}$ in the figure (median overall rSD at N=15 - should be median FYI rSD at N=15). This has been corrected, and the "combined" plot has been removed as it is no longer necessary. Lines 220-223 have been updated to reflect why we chose to do this.*

255-261, Figure 9: There is really a significant difference near the ice edge – e.g., the Odden feature. It makes sense that it is related to thin/new ice, but I'm surprised by how large it is. Since, NIC used ESMR, perhaps it is due to how the analysts assessed the TB values back in the 1970s – maybe the manual analysis effectively used a lower TB threshold? Another possibility is simply that analysts were being very conservative in where they drew the ice edge. This has been an issue in other comparisons – the analysts want to err on the side of too much ice vs. missing ice. In other words, the fact that that V1.1 shows less ice than NIC may not be a low bias by V1.1, but a high bias by NIC.

*Reply: We agree that the ice charts could be overly conservative and that might explain some of the differences between ESMR and NIC SIC. We have indicated that possibility in the discussion.*

Figure 9: Another add feature I see is that that while V1.1 is low relative to NIC right near the ice edge, it appears that in several regions (e.g., Bering Sea), beyond that low bias area, there is a "rim" of red shades, indicating a high bias in V1.1. How does this happen? Presumably this is well into open water.

*Reply: This is a good question and we do not have a definite answer. But we think that actually the ice edge in the ice chart and in the ESMR SIC is aligned pretty well. However, because of the coarse resolution of ESMR there will be some smearing so that in open water near the ice edge there will be higher ESMR SIC than the ice chart SIC (which is correctly*

*zero). Over ice near the ice edge there are two things: 1) the ice chart SIC is probably overestimated, and 2) The ESMR SIC will underestimate SIC near the edge due to smearing and due to the presence of new-ice near the edge. We have expanded the discussion of this topic in the text.*

256: missing ")" in the list of references.

*Reply: Thanks.*

---

## Referee Report (RR1)

The authors have addressed the comments and concerns raised in the previous reviews, including incorporating the additional arguments into the manuscript. The paper is now clearer in outlining the consistency of the methodology applied to both hemispheres, as well as the differences in the validation of the dataset for the Arctic and Antarctic regions. The value of the dataset for both regions is also better highlighted.

Overall, the manuscript has improved in the aspects noted in the previous reviews, and the figures are clearer. Only a few minor typographical issues remain (listed below). Otherwise, I believe the manuscript is suitable for publication.

Typo:

Line 150: ": The filter". Replace the colon with a full stop.
Line 217: For clarity, consider revising to: "for each cell across all odd values".
Caption Fig. 4: "Boxplots illustrates" → change to "Boxplots illustrate".